# Diagnostic Utility of CT Findings as Indicators for Bowel Resection in Strangulated Small Bowel Obstruction

**DOI:** 10.3390/jcm14228027

**Published:** 2025-11-12

**Authors:** Takashi Okumura, Shingo Tsujinaka, Nozomi Satani, Kuniharu Yamamoto, Toru Nakano, Takayuki Yamada, Yu Katayose, Chikashi Shibata

**Affiliations:** 1Department of Surgery, Division of Gastroenterologic Surgery, Tohoku Medical and Pharmaceutical University, Sendai 983-8536, Miyagi, Japan; 2Division of Radiology, Tohoku Medical and Pharmaceutical University, Sendai 983-8536, Miyagi, Japan; 3Department of Surgery, Division of Hepato-Biliary and Pancreatic Surgery, Tohoku Medical and Pharmaceutical University, Sendai 983-8536, Miyagi, Japan

**Keywords:** bowel resection, bowel wall enhancement, computed tomography, indocyanine green fluorescence, strangulated small bowel obstruction

## Abstract

**Background/objectives:** Strangulated small bowel obstruction (SSBO) is a life-threatening condition that often requires emergency surgery. Identifying preoperative computed tomography (CT) findings indicative of bowel resection may improve diagnostic accuracy and inform surgical decision-making. **Methods:** We retrospectively analyzed patients diagnosed with SSBO who underwent contrast-enhanced abdominal CT and emergency surgery between January 2022 and April 2024. Patients were divided into two groups according to the surgical outcomes: those who underwent bowel resection and those who did not. CT images were independently reviewed by a radiologist blinded to surgical outcomes, and CT findings were compared between the resection and non-resection groups. Variables significant in the between-group comparisons (*p* < 0.05) were entered into a multivariable logistic regression to identify indicators for bowel resection. **Results:** Fifty-two patients were identified, sixteen (30.8%) of whom required bowel resection. The most reliable indicator was absent bowel wall enhancement on contrast-enhanced CT, with a sensitivity of 75.0% and specificity of 86.1%. It was also independently associated with bowel resection [odds ratio (OR) 19.7; 95% confidence interval: 3.43–113.4]. In contrast, ascites, beak sign, and mesenteric edema were commonly observed in both groups and lacked specificity. Of note, bowel resection was avoided in 5 of 17 patients with absent bowel wall enhancement based on intraoperative assessment using indocyanine green (ICG) fluorescence imaging. **Conclusions:** Absent bowel wall enhancement on contrast-enhanced CT is an independent preoperative indicator for bowel resection in SSBO.

## 1. Introduction

Strangulated small bowel obstruction (SSBO) accounts for approximately 16–18% of all bowel obstructions and represents a rapidly progressive condition that, if not promptly treated, can lead to bowel ischemia, necrosis, septic shock, and death [1,2]. Although elevated lactate levels and metabolic acidosis may suggest the presence of bowel necrosis, these laboratory abnormalities are often absent in the early stages of an obstruction, making timely diagnosis difficult [3,4,5]. In this context, abdominal computed tomography (CT) plays a central role in the diagnosis and evaluation of SSBO by providing crucial information on both the presence and severity of the obstruction [6,7,8]. While surgery is generally required for SSBO, the decision to perform bowel resection depends on an intraoperative assessment of bowel viability; therefore, identifying the CT findings that predict the need for resection in the preoperative setting is of significant clinical importance. Early identification of such findings can support surgical planning, inform practitioners on the level of urgency, and potentially improve patient outcomes. Various CT signs, such as the beak sign and closed loop, have been proposed as indicators of strangulation or ischemia [9,10]. However, their diagnostic accuracy varies considerably, and previous studies often lacked standardized inclusion criteria or failed to clearly differentiate between patients who underwent bowel resection and those who did not [11,12,13]. Furthermore, interobserver variability and inconsistent definitions of bowel viability further complicate interpretation and clinical application [14,15]. Given the emergency nature of SSBO, it is essential that radiologists and surgeons, in addition to other frontline clinicians, recognize which CT features most reliably indicate the need for bowel resection, rather than merely confirm obstruction. In this study, we aim to evaluate the diagnostic utility of various preoperative CT findings as indicators for bowel resection in patients with SSBO and identify the imaging features most helpful in guiding urgent surgical management.

## 2. Materials and Methods

### 2.1. Study Design and Patients

This retrospective observational study included patients who underwent surgery at our institution with diagnosis of SSBO between 1 January 2022, and 30 April 2024, and was approved by the Research Ethics Committee for Life Science and Medical Research, Tohoku Medical and Pharmaceutical University (2024-2-046). Eligible patients met all of the inclusion criteria and none of the exclusion criteria.

#### 2.1.1. The Inclusion Criteria Were as Follows

Diagnosis of SSBO based on clinical and radiological findings.Implementation of preoperative contrast-enhanced abdominal CT.Surgical intervention during the same hospital admission.Age ≥ 20 years.

#### 2.1.2. The Exclusion Criteria Were as Follows

1.Patients who improved with non-surgical conservative treatment alone.2.Absence of contrast-enhanced CT.3.Incomplete clinical or imaging data.4.Patients who underwent surgery more than 24 h after contrast-enhanced CT and in whom there were no suspicions of SSBO at the time of surgical decision.

### 2.2. CT Protocol

All examinations were performed using an 80-detector-row CT scanner (Aquilion Prime SP i; Canon Medical Systems, Otawara, Tochigi, Japan). The scanning parameters were as follows: collimation of 0.5 × 80, pitch of 0.813, slice thickness of 1.0 mm, and gantry rotation time of 0.5 s. Contrast-enhanced scans were obtained after intravenous administration of iodinated contrast medium using a mechanical injector. Details of acquisition protocols are summarized in the (Table A1, Appendix A), including the combinations of tube voltage, iodine dose, and scan delay.

### 2.3. Preoperative CT Assessment by Radiologist

All preoperative CT scans were retrospectively reviewed by a board-certified radiologist with more than 15 years of experience who was blinded to surgical outcomes. CT images were obtained directly from the institutional imaging archive system without modification; were evaluated in the axial, coronal, and sagittal planes; and all contrast phases were reviewed. The radiologist assessed the presence of the findings which were suggestive of SSBO based on the predefined diagnostic criteria established in the previous literature [1,8,10,16], with attention paid to morphologic changes and signs of vascular compromise, as outlined in the next section.

### 2.4. CT Findings Suggestive of SSBO

Representative images are presented in Figure 1. 

Closed loop (Figure 1a): A dilated U- or C-shaped bowel segment with two transition points in close proximity, often associated with radial distribution of mesenteric vessels. This configuration may carry a high risk of rapid progression to strangulation and ischemia, requiring prompt surgical consideration.Beak sign (Figure 1b): Gradual tapering of the bowel lumen or contrast column at the obstruction point, typically indicating torsion or sharp angulation. This sign localizes the obstruction and often reflects mechanical blockage due to adhesions or volvulus, which may progress to ischemia if left untreated.Whirl sign (Figure 1c): Swirling of mesenteric vessels and fat, suggestive of volvulus or twisted mesentery. Presence of this sign strongly suggests torsion with compromised mesenteric blood flow.Small bowel feces sign (Figure 1d): Mixture of gas and particulate matter (resembling feces) within a dilated small bowel loop proximal to the obstruction, indicating delayed transit. This sign indicates subacute or prolonged obstruction, but does not necessarily imply ischemia.Mesenteric edema (Figure 1e): Increased attenuation and stranding of mesenteric fat surrounding the affected loop, typically reflecting venous congestion. This is an early indicator of impaired venous outflow and increased risk of ischemia, requiring close monitoring and a low threshold for surgical intervention if clinical deterioration occurs.Mesenteric vessel engorgement (Figure 1f): Prominent or dilated mesenteric veins near the involved segment, suggestive of impaired venous outflow or strangulation. Venous congestion typically precedes arterial compromise, and the presence of this sign indicates evolving mesenteric ischemia.Absent bowel wall enhancement (Figure 1g): Complete absence of bowel wall enhancement on contrast-enhanced CT, which is highly suggestive of transmural infarction. This is a critical sign of irreversible ischemia and generally indicates the need for immediate surgical intervention.Blurred Kerckring folds (Figure 1h): Indistinct mucosal folds (valvulae conniventes) in dilated small bowel loops, often associated with ischemic edema. Blurring of mucosal folds reflects mucosal/submucosal injury from ischemia and impending bowel damage due to severe obstruction.Ascites (Figure 1i): Free peritoneal fluid, either localized around the affected loop or diffusely distributed, is frequently associated with advanced ischemia. Increasing ascites in the obstruction setting often indicates transmural ischemia or severe inflammation.

**Figure 1 jcm-14-08027-f001:**
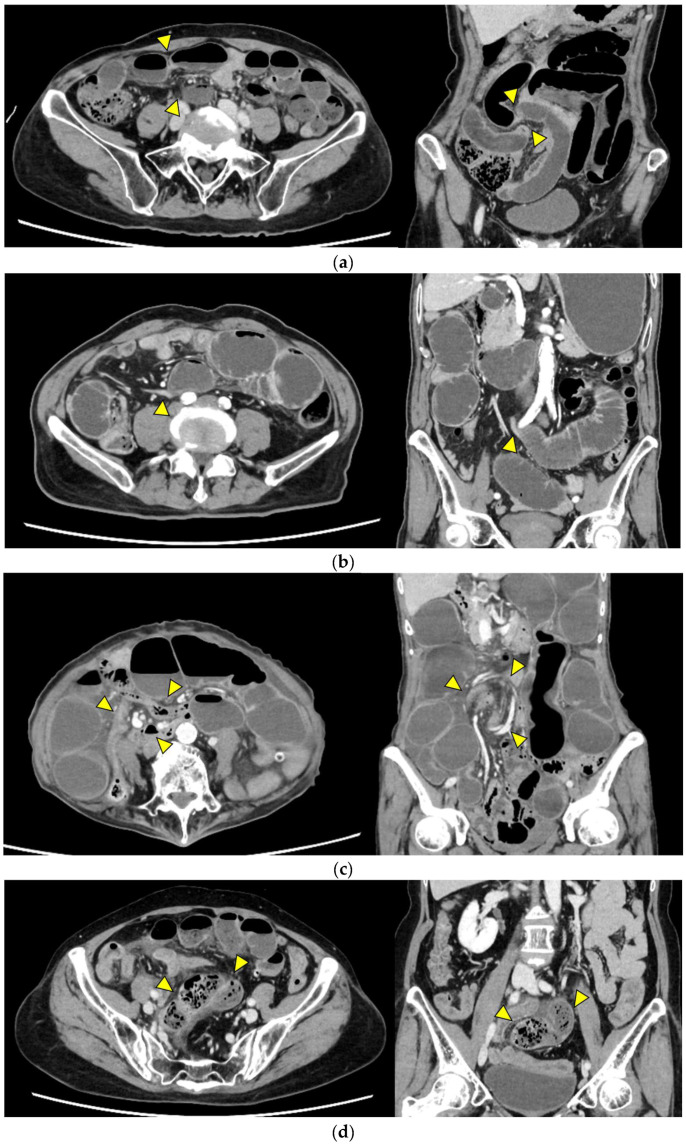
Representative CT findings suggestive of SSBO. Yellow arrowheads highlight the relevant findings. (**a**) Closed loop; (**b**) beak sign; (**c**) whirl sign; (**d**) small bowel feces sign; (**e**) mesenteric edema; (**f**) mesenteric vessel engorgement; (**g**) absent bowel wall enhancement; (**h**) blurred Kerckring folds; (**i**) ascites.

### 2.5. Surgical Procedure

All patients underwent emergency surgery for SSBO based on clinical and radiologic evaluation. The primary outcome of this study was the need for bowel resection; according to operative records, patients were classified into either the resection or non-resection group based on intraoperative findings. The decision to perform bowel resection was made by the operating surgeon, primarily according to gross evidence of irreversible ischemia. In some cases, ICG fluorescence imaging was used intraoperatively to assess bowel perfusion; if perfusion was deemed sufficient, resection was avoided. The use of ICG in this context has been reported to reduce unnecessary bowel resection in SSBO [17] and is recommended by international consensus guidelines for emergency surgery [18]. Final decisions regarding resection were made at the discretion of the attending surgeon.

### 2.6. Study Parameters

Patient-related variables included age, sex, body mass index (BMI), American Society of Anesthesiologists physical status (ASA-PS), comorbidities, history of abdominal surgery, and etiology of SSBO. The primary outcome was whether or not bowel resection was performed during surgery, while operative findings, including the reasons for making the decision regarding bowel resection, were extracted from surgical reports documented in the medical records. The interval between contrast-enhanced CT and the surgery start time was recorded for each patient and compared between the resection and non-resection groups.

### 2.7. Statistical Analysis

Categorical variables of baseline characteristics were presented as frequencies and percentages, and age was expressed as the median (interquartile range). Comparisons between the resection and non-resection groups were performed using the Mann–Whitney U test for age and the Chi-square or Fisher’s exact test for categorical variables, as appropriate. The diagnostic performance of each CT finding was first evaluated by calculating sensitivity and specificity from 2 × 2 contingency tables, using intraoperative findings as the reference standard. CT finding frequencies were compared between groups using the Chi-square or Fisher’s exact test, while mutilcollinearity among CT findings was assessed using the variance inflation factor (VIF), with a value greater than 5 considered to indicate multicollinearity. Thereafter, variables with a *p*-value < 0.05 were entered into a multivariate logistic regression model to determine independent indicators. For the independent predictors identified in the multivariate analysis, receiver operating characteristic (ROC) curve analysis was conducted and the area under the curve (AUC) was calculated. Statistical significance was set at *p* < 0.05. All analyses were performed using SPSS Statistics, version 25 (IBM Corp., Armonk, NY, USA).

## 3. Results

### 3.1. Patient Characteristics

Eighty-three patients were screened for eligibility, of which thirty-one were excluded due to factors including undergoing surgery 24 h after contrast-enhanced CT without suspicion of SSBO at the time of surgical decision (*n* = 9), undergoing emergency surgery without contrast-enhanced CT (*n* = 21), and having unevaluable CT results due to the enteric contrast study (*n* = 1). Ultimately, fifty-two patients met the inclusion criteria and were included in the analysis. The patient selection process is illustrated in Figure 2, while patient characteristics are summarized in Table 1. There were no significant differences between the resection (*n* = 16) and non-resection groups (*n* = 36) in terms of sex distribution (*p* = 0.77), age (*p* = 0.36), or BMI (*p* = 0.41). The proportion of patients with ASA-PS class 3–4 tended to be higher in the resection group, but the difference was not statistically significant (*p* = 0.11). A history of abdominal surgery was common in both groups (*p* = 0.83), and the prevalence of cardiac disease (*p* = 0.80), hypertension (*p* = 0.55), cerebrovascular disease (*p* = 0.58), and diabetes mellitus (*p* = 0.70) was similar between the two groups.

### 3.2. Etiology of SSBO

The etiologies of SSBO are summarized in Table 2. In both groups, the most common etiology of SSBO was adhesion, present in 11 patients (68.8%) in the resection group and 28 (77.8%) in the non-resection group. Other etiologies in the resection group included an incarcerated abdominal wall hernia, inguinal hernia, and intussusception due to cecal cancer. In the non-resection group, additional etiologies included various types of incarcerated hernias (inguinal, femoral, scar, abdominal wall), intestinal volvulus, and internal hernia.

### 3.3. Diagnostic Performance of CT Findings

The diagnostic performance of preoperative CT findings is shown in Table 3. In patients who underwent bowel resection, the highest sensitivities were shown for ascites and beak sign (both 87.5%, 95% CI 64.0–96.5), followed by absent bowel wall enhancement and mesenteric edema (both 75.0%, 95% CI 50.5–89.8), and finally closed loop (68.8%, 95% CI 44.4–85.8). The highest specificity value was shown for mesenteric vessel engorgement (94.4%, 95% CI 81.9–98.5), followed by absent bowel wall enhancement (86.1%, 95% CI 71.3–93.9); whirl sign, blurred Kerckring folds, and small bowel feces sign (all 83.3%, 95% CI 68.1–92.1); and finally closed loop (61.1%, 95% CI 44.9–75.2).

### 3.4. Frequency of Preoperative CT Findings Between Groups

The CT finding frequencies are summarized in Table 4. Absent bowel wall enhancement was significantly more frequent in the resection (12/16) than in the non-resection group (5/36, *p* < 0.001), and closed loop was also significantly more common in the resection group (11/16 vs. 14/36, *p* = 0.047).

Mesenteric vessel engorgement tended to be more frequent in the resection group (4/16 vs. 2/36), although the difference did not reach statistical significance (*p* = 0.064). In contrast, ascites, beak sign, and mesenteric edema were prevalent in both groups with no statistically significant differences.

### 3.5. Multicollinearity Assessment

A multicollinearity assessment revealed that all VIF values were below 5. Among the CT findings, mesenteric edema had the highest VIF (2.315), followed by closed loop (2.18), absent bowel wall enhancement (1.762), ascites (1.738), blurred Kerckring folds (1.514), beak sign (1.397), whirl sign (1.283), mesenteric vessel engorgement (1.174), and small bowel feces sign (1.171), indicating no significant multicollinearity.

### 3.6. Multivariate Logistic Regression Analysis

The results of the multivariate analysis are shown in Table 5. Variables with a *p*-value < 0.05 in the between-group frequency comparisons of preoperative CT findings were entered into the multivariate model, after which absent bowel wall enhancement remained the only independent indicator of bowel resection (OR 19.71, 95% CI 3.43–113.4, *p* = 0.001); closed loop (OR 0.895, 95% CI 0.15–5.22, *p* = 0.902) was not significant.

### 3.7. ROC Curve Analysis

The ROC curve analysis is shown in Figure 3. For predicting bowel resection, absent bowel wall enhancement yielded a sensitivity and specificity of 75.0% and 86.1%, respectively, and had an AUC of 0.806.

### 3.8. Exceptions in Imaging–Surgical Correlations

Among the seventeen patients with absent bowel wall enhancement, five did not undergo bowel resection. In four of these cases, intraoperative assessment using ICG fluorescence angiography demonstrated sufficient perfusion after a reduction in the strangulated segment, allowing for the bowel to be preserved (Figure 4a,b). In the remaining case involving an incarcerated incisional hernia, no signs of irreversible ischemia were observed during surgery, and resection was not required despite the imaging findings.

Conversely, four patients underwent bowel resection despite having no absence of bowel wall enhancement. One case involved strong adhesions from an incarcerated incisional hernia requiring resection regardless of perfusion status, while another patient had intussusception due to a cecal carcinoma, for which bowel resection was indicated for oncologic purposes. In the remaining two cases, surgery was delayed by 2 to 4 h after CT imaging, and so ischemia may have progressed beyond what was initially detectable.

### 3.9. Interval Between CT and Surgery

The interval between CT and surgery ranged from 2 to 23 h (median: 3.4 h) (Table 6), with a shorter median interval in the resection group (2.9 h; range: 2–6 h) than in the non-resection group (4.1 h; range: 2–23 h).

## 4. Discussion

### 4.1. Overview of Key Findings

In this study, we examined the preoperative CT findings associated with the need for bowel resection in patients with SSBO. Among the evaluated radiologic features, absent bowel wall enhancement was the most reliable indicator, demonstrating both high specificity (86.1%) and reasonable sensitivity (75.0%). Multivariate analysis confirmed this finding as an independent indicator, with an odds ratio of 19.7. These results are consistent with previous studies reporting that absent bowel wall enhancement on contrast-enhanced CT correlates strongly with transmural necrosis [6,10,13,14].

### 4.2. Clinical Significance of Other CT Findings

Closed loop was also associated with bowel resection in univariate but not multivariate analysis, suggesting that it does not inevitably lead to bowel necrosis. This configuration represents a mechanical condition that predisposes the bowel to strangulation, but the actual development of ischemia depends on additional factors such as the severity and duration of vascular compromise, collateral circulation, and the timing of surgical intervention. In our study, some patients with closed loop may have undergone timely surgery before irreversible ischemic changes occurred, which may explain the lack of independent association. The results of previous studies have similarly shown that closed loop reflects a predisposing morphology for strangulation, rather than a direct indicator of transmural necrosis [19,20,21].

CT signs, such as beak sign, ascites, and mesenteric edema, were frequently observed in both groups, limiting their specificity. These signs can also appear in non-strangulated obstruction due to venous congestion or increased permeability from simple obstruction. Previous reports have shown that mesenteric or peritoneal fluid accumulation reflect disease severity, but do not necessarily indicate necrosis [22,23]; however, these features may still aid in early recognition of SSBO, particularly when interpreted alongside clinical and laboratory findings.

### 4.3. Discrepancies Between Intraoperative Decisions and CT Findings

Several noteworthy exceptions were identified regarding the diagnostic accuracy of absent bowel wall enhancement as an indicator for bowel resection. Among the seventeen patients with this finding, five did not require resection; in four of these cases, intraoperative ICG fluorescence angiography demonstrated adequate perfusion following reduction in the strangulated loop, allowing for bowel preservation. In our series, ICG fluorescence was selectively applied when intraoperative bowel viability remained uncertain after reduction. These observations are consistent with previous reports showing that ICG fluorescence can help prevent unnecessary resections [17,18].

In contrast, four patients required bowel resection despite having preserved bowel wall enhancement. These exceptions highlight two important considerations: First, preserved enhancement on contrast-enhanced CT does not necessarily exclude evolving ischemia. In early or venous-predominant ischemia, or following transient reperfusion, apparently viable enhancement can still progress rapidly to transmural necrosis [24,25]. CT should therefore be regarded as a snapshot within a dynamic disease process. Second, clinical context may override imaging findings in determining the need for resection; incarcerated hernias with dense adhesions may necessitate resection due to irreversible injury even when enhancement is preserved. Similarly, adult colonic-type intussusception is frequently malignant, and oncologic resection is generally indicated [26,27].

Taken together, our results indicate that absent bowel wall enhancement remains the strongest imaging indicator for bowel resection in SSBO, supporting the routine use of contrast-enhanced CT in preoperative evaluation. ICG fluorescence may serve as a valuable intraoperative adjunct, particularly in cases with ambiguous bowel viability, avoiding unnecessary resections. Nonetheless, clinical judgment should remain paramount, as imaging findings alone may not fully capture the intraoperative pathology, especially in delayed or complex presentations.

### 4.4. Limitations

#### 4.4.1. Study Design and Sample Size

The retrospective and single-center design may limit the generalizability of these findings. In addition, the sample size was relatively small, which may have affected statistical significance, particularly in subgroup analyses.

#### 4.4.2. Interobserver Variability

Radiologic assessments were performed by a single radiologist, which may have led to interpretation bias. Future studies should incorporate interobserver agreement analysis to validate these findings.

#### 4.4.3. CT Acquisition Protocols

CT acquisition protocols were not entirely uniform across patients, particularly with respect to tube voltage, iodine dose, and scan delay. However, corresponding combinations of tube voltages and iodine doses were theoretically equivalent in CT attenuation values, and the distribution of protocols was broadly comparable between the resection and non-resection groups; therefore, these differences were unlikely to have materially influenced the study outcomes.

#### 4.4.4. Assessment of Bowel Wall Enhancement

Bowel wall enhancement on contrast-enhanced CT was assessed in a binary manner, determined as being either present or absent. However, this dichotomous classification does not fully account for the multiple factors influencing enhancement, including the aforementioned CT protocols. Ideally, a graded or quantitative scoring system would allow for a more precise evaluation of enhancement patterns. Nevertheless, given the clinical urgency and need for critical surgical decision-making in SSBO, we deliberately adopted an absolute criterion to identify CT findings that could strongly support the indication for bowel resection.

#### 4.4.5. Interval Between CT and Surgery

The CT-to-surgery interval varied considerably among patients (2–23 h) and was significantly shorter in the bowel resection group compared to the non-resection group. This partly reflects the prompt surgical decision-making in patient with CT findings highly suggestive of bowel ischemia. However, the timing of surgery influenced not only by clinical prioritization, but also by logistical factors such as operating room availability, staff scheduling (surgeons, anesthesiologists, nurses, and clinical engineers), and patient-related considerations including informed consent and family discussions. These differences in the timing of interventions may have contributed to discrepancies between imaging findings and intraoperative outcomes. This variability should be considered when interpreting imaging findings in clinical practice.

#### 4.4.6. Use of ICG Fluorescence Imaging

ICG fluorescence imaging was not uniformly applied across all cases and was selectively used at the discretion of the operating surgeon, which could have introduced selection bias in the evaluation of bowel viability.

## 5. Conclusions

Absent bowel wall enhancement on contrast-enhanced CT images was identified as the most reliable preoperative indicator for bowel resection in patients with SSBO. Given its high diagnostic performance, contrast-enhanced CT should be routinely utilized in clinical evaluation; however, radiologic findings should be interpreted in conjunction with clinical presentation and intraoperative assessment, as the diagnosis and management of SSBO require a comprehensive evaluation rather than imaging alone. Integrating CT features with surgical and clinical judgment is essential to optimize decision-making and improve patient outcomes.

## Figures and Tables

**Figure 2 jcm-14-08027-f002:**
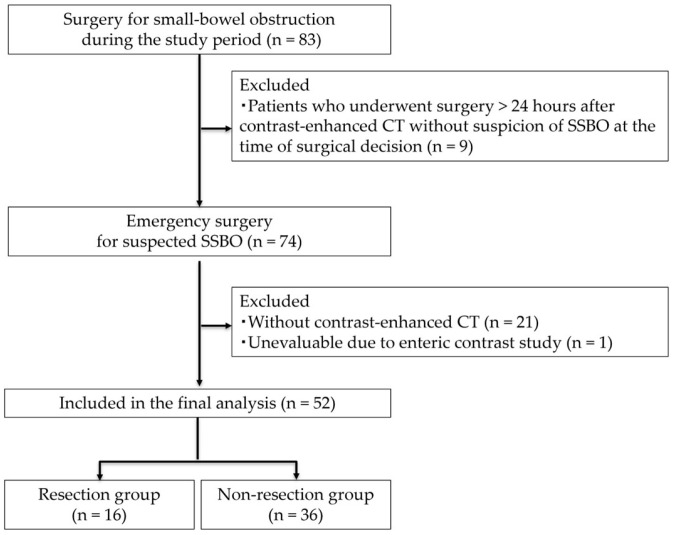
Flow diagram of patient selection with inclusion and exclusion criteria.

**Figure 3 jcm-14-08027-f003:**
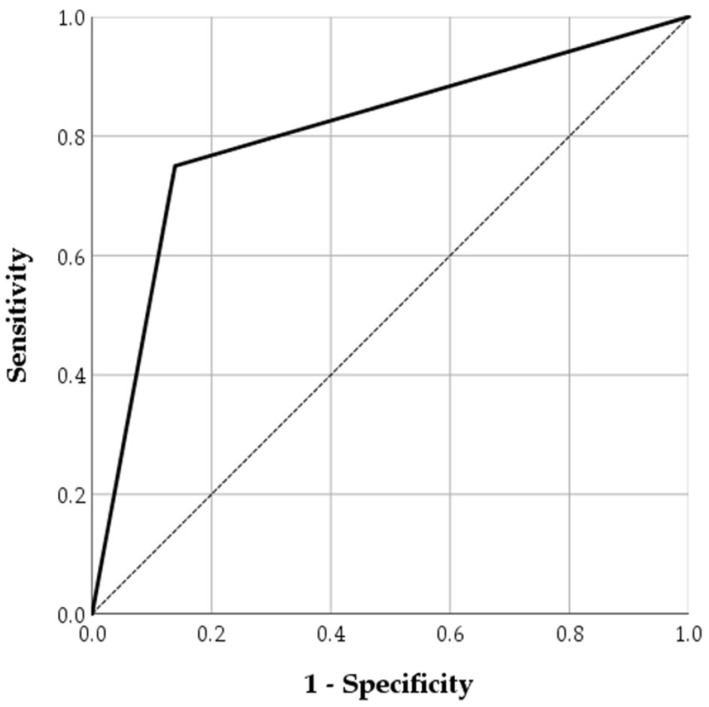
Receiver operating characteristic curve for absent bowel wall enhancement in predicting bowel resection in SSBO. The dotted diagonal line represents the reference line for random classification (i.e., AUC = 0.5).

**Figure 4 jcm-14-08027-f004:**
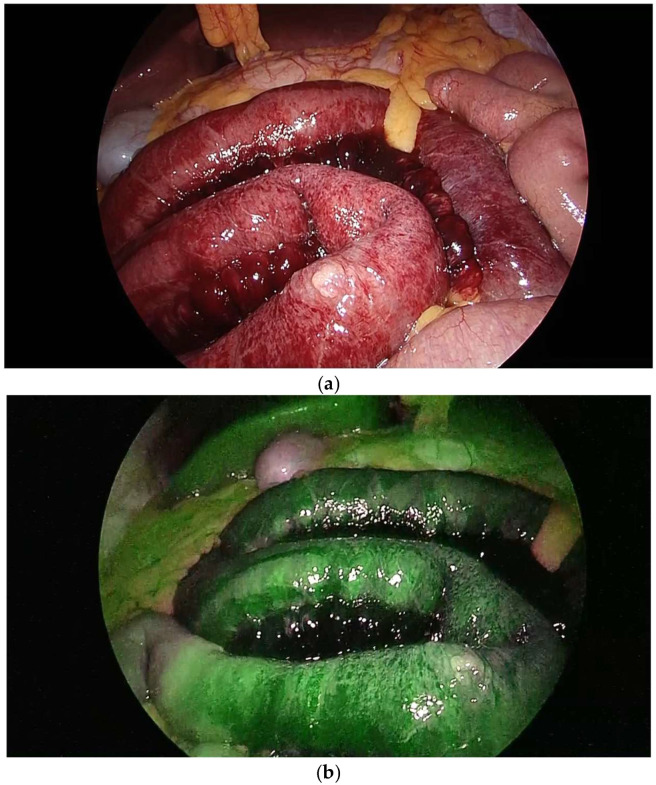
Representative images of intraoperative assessment using indocyanine green (ICG) fluorescence angiography (all images were obtained directly from the institutional imaging archive system without modification). (**a**) Closed loop before ICG assessment: congestion of the bowel wall and hematoma in the mesentery are observed; (**b**) ICG assessment: diffuse distribution of ICG to the bowel wall proves sufficient vascularization.

**Table 1 jcm-14-08027-t001:** Baseline characteristics of patients in the resection and non-resection groups.

Characteristics	Resection Group (*n* = 16)	Non-Resection Group (*n* = 36)	*p*-Value
Male	6 (37.5)	12 (33.3)	0.77
Age (years)	77.5 (49–96)	75 (20–92)	0.36
BMI (kg/m^2^)	19.1 (15.8–31.4)	20.7 (15.3–32.5)	0.41
ASA-PS: III and IV	7 (43.8)	8 (22.2)	0.11
History of abdominal surgery	12 (75.0)	26 (72.2)	0.83
* Cardiac disease	5 (31.3)	10 (27.8)	0.80
Hypertension	9 (56.3)	17 (47.2)	0.55
** Cerebrovascular disease	1 (6.2)	4 (11.1)	0.58
Diabetes mellitus	2 (12.5)	6 (16.7)	0.70

Data are presented as number (%) or median (range). * Cardiac disease includes ischemic heart disease, heart failure, and arrhythmia. ** Cerebrovascular disease includes ischemic or hemorrhagic stroke. BMI: body mass index; ASA-PS: American Society of Anesthesiologists physical status.

**Table 2 jcm-14-08027-t002:** Etiologies of SSBO in the resection and non-resection groups.

Etiologies	Resection Group (*n* = 16)	Non-Resection Group (*n* = 36)
Adhesion	11 (68.8)	28 (77.8)
Incarcerated ventral hernia	1 (6.3)	1 (2.8)
Intussusception (due to cecal cancer)	1 (6.3)	0
Incarcerated inguinal hernia	2 (12.5)	3 (8.3)
Incarcerated femoral hernia	1 (6.3)	2 (5.6)
Intestinal volvulus	0	1 (2.8)
Internal hernia	0	1 (2.8)

Data are presented as number (%).

**Table 3 jcm-14-08027-t003:** Diagnostic performance of preoperative CT findings for predicting bowel resection.

CT Findings	Sensitivity (%) (95% CI)	Specificity (%) (95% CI)
Closed loop	68.8 (44.4–85.8)	61.1 (44.9–75.2)
Beak sign	87.5 (64.0–96.5)	13.9 (6.1–28.7)
Whirl sign	12.5 (3.5–36.0)	83.3 (68.1–92.1)
Small bowel feces sign	12.5 (3.5–36.0)	83.3 (68.1–92.1)
Mesenteric edema	75.0 (50.5–89.8)	30.6 (18.0–46.9)
Mesenteric vessel engorgement	25.0 (10.2–49.5)	94.4 (81.9–98.5)
Absent bowel wall enhancement	75.0 (50.5–89.8)	86.1 (71.3–93.9)
Blurred Kerckring folds	25.0 (10.2–49.5)	83.3 (68.1–92.1)
Ascites	87.5 (64.0–96.5)	16.7 (7.9–31.9)

**Table 4 jcm-14-08027-t004:** Frequency of preoperative CT findings in the resection and non-resection groups.

CT Findings	All Cases (*n* = 52)	Resection Group (*n* = 16)	Non-Resection Group (*n* = 36)	*p*-Value
Closed loop	25	11 (68.8)	14 (38.9)	0.047
Beak sign	45	14 (87.5)	31 (86.1)	1.0
Whirl sign	8	2 (12.5)	6 (16.7)	1.0
Small bowel feces sign	8	2 (12.5)	6 (16.7)	1.0
Mesenteric edema	37	12 (75.0)	25 (69.4)	0.75
Mesenteric vessel engorgement	6	4 (25.0)	2 (5.6)	0.064
Absent bowel wall enhancement	17	12 (75.0)	5 (13.9)	<0.001
Blurred Kerckring folds	10	4 (25.0)	6 (16.7)	0.48
Ascites	44	14 (87.5)	30 (83.3)	1.0

Data are presented as number (%).

**Table 5 jcm-14-08027-t005:** Multiple binary logistic regression for risk factors for bowel resection.

	B	S.E.	Wald	OR	95%CI	*p*-Value
Closed loop	−0.111	0.900	0.015	0.895	0.153–5.22	0.902
Absent bowel wall enhancement	2.981	0.893	11.152	19.71	3.426–113.377	0.001

**Table 6 jcm-14-08027-t006:** Interval between contrast-enhanced CT and surgery.

	*n*	Median (h)	Range (h)	*p*-Value
All cases	52	3.4	2.0–23.3	
Non-resection	36	4.1	2.0–23.3	0.004
Resection	16	2.9	2.0–5.8	

## Data Availability

The data supporting the findings of this study are available from the corresponding author on reasonable request.

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
