# Peer review of "J. Clin. Med.2025, 14(22), 8027;https://doi.org/10.3390/jcm14228027"

_jcm, 2025, doi:10.3390/jcm14228027_

Round 1

Reviewer 1 Report

Comments and Suggestions for Authors

This manuscript presents a retrospective analysis identifying preoperative CT findings predictive of bowel resection in patients with SSBO. The study demonstrates that absent bowel wall enhancement is the strongest independent indicator, while other CT signs show limited specificity. The work is clinically relevant, clearly presented, and methodologically sound, although clarification regarding CT protocols, timing of surgery, and interobserver variability is recommended. 

Strengths

  • The study tackles a clinically important question with direct implications for emergency surgical care.
  • Methodology is appropriate, with predefined CT criteria and blinded radiologic assessment.
  • Results are clearly presented, identifying absent bowel wall enhancement as the strongest predictor for resection, supported by statistical analysis.
  • The discussion integrates findings coherently with previous literature and appropriately emphasizes the complementary role of intraoperative ICG fluorescence.

Weaknesses / Points for Revision

  1. Study population and flow: Clarify the total number of SSBO cases initially screened, exclusions, and reasons for exclusion. A simple flow diagram would improve transparency.
  2. CT protocol variability: Address the heterogeneity in acquisition parameters (tube voltage, iodine dose, scan delay). Indicate whether any attenuation correction or standardization was applied.
  3. Statistical reporting: Provide confidence intervals for sensitivity, specificity, and odds ratios. Confirm whether multicollinearity among CT variables was checked prior to regression analysis.
  4. Timing of intervention: Include data on the interval between CT and surgery, as delays may influence imaging–operative correlation.
  5. Quantitative analysis: If feasible, include a ROC analysis to demonstrate the diagnostic performance (AUC) of absent bowel wall enhancement relative to other CT findings.
  6. Interobserver variability: Acknowledge the limitation of single-reader interpretation and suggest validation through interobserver agreement in future studies.
  7. Discussion clarity: Briefly explore why other expected CT indicators (mesenteric edema, ascites) were not statistically significant. The limitations section should be condensed and organized thematically for easier reading.

Other Comments

  • Correct minor typographical issues (e.g., “forth” → “fourth”), and ensure consistency in hyphenation (e.g., “preoperative” throughout).
  • Verify reference style compliance with J. Clin. Med. formatting.
  • Figures and legends are clear, but consider adding brief clinical context for educational value.
  • The conclusion should stress integration of radiologic findings with clinical and intraoperative assessment, rather than relying on imaging alone.

Author Response

This manuscript presents a retrospective analysis identifying preoperative CT findings predictive of bowel resection in patients with SSBO. The study demonstrates that absent bowel wall enhancement is the strongest independent indicator, while other CT signs show limited specificity. The work is clinically relevant, clearly presented, and methodologically sound, although clarification regarding CT protocols, timing of surgery, and interobserver variability is recommended. 

Strengths

  • The study tackles a clinically important question with direct implications for emergency surgical care.
  • Methodology is appropriate, with predefined CT criteria and blinded radiologic assessment.
  • Results are clearly presented, identifying absent bowel wall enhancement as the strongest predictor for resection, supported by statistical analysis.
  • The discussion integrates findings coherently with previous literature and appropriately emphasizes the complementary role of intraoperative ICG fluorescence.

Response: Thank you for highlighting the strengths and weaknesses regarding out manuscript. In response to your comment and Reviewer 2’s related suggestion, we have refined the patient selection criteria to exclude cases where surgery was performed more than 24 hours after contrast-enhanced CT and SSBO was not suspected at the time of surgical decision. This adjustment ensures a more homogeneous study population and minimizes potential bias related to ischemia progression over time.

As a result of this stricter criterion, the number of included cases has slightly changed. While the overall conclusions remain consistent with the original analysis, some data points have been updated accordingly. We have transparently reflected these changes in the Methods, Results, and Discussion sections.

Weaknesses / Points for Revision

1. Study population and flow: Clarify the total number of SSBO cases initially screened, exclusions, and reasons for exclusions. A simple flow diagram would improve transparency.

Response: We appreciate your suggestion. We have clarified the total number of cases screened, exclusions, and reasons for exclusion in the Results section. Additionally, a flow diagram illustrating the patient selection process has been added as Figure 2.

2. CT protocol variability: Address the heterogeneity in acquisition parameters (tube voltage, iodine dose, scan delay). Indicate whether any attenuation correction or standardization was applied.

Response: Thank you for your comment. We have expanded the Appendix to provide detailed information on CT acquisition parameters, including tube voltage, iodine dose, and scan delay. We have also clarified that injection timing and iodine concentration were standardized according to institutional protocol, while tube voltage was adjusted based on body habitus. No attenuation correction or image normalization was applied, as all CT findings were visually assessed using a consistent interpretive approach.

3. Statistical reporting: Provide confidence intervals for sensitivity, specificity, and odds ratios. Confirm whether multicollinearity among CT variables was checked prior to regression analysis.

Response: Thank you for your valuable comments. We have added 95% confidence intervals for sensitivity, specificity, and odds ratios in the Results section. In addition, we assessed multicollinearity among CT variables using the variance inflation factor (VIF) prior to regression analysis. The maximum VIF was 2.315, indicating no significant multicollinearity. The description has been added to the Methods and Results section.

4. Timing of intervention: Include data on the interval between CT and surgery, as delays may influence imaging–operative correlation.

Response: Thank you for your suggestion. We have added data on the interval between CT and surgery in the Methods and Results sections (including median, range, and group comparison) and discussed its potential impact on imaging–operative correlation in the Discussion section.

5. Quantitative analysis: If feasible, include a ROC analysis to demonstrate the diagnostic performance (AUC) of absent bowel wall enhancement relative to other CT findings.

Response: Thank you for your suggestion. We have conducted ROC curve analysis to demonstrate the diagnostic performance of absent bowel wall enhancement relative to other CT findings. The results, including AUC, sensitivity, and specificity, are presented in the Results section and illustrated in Figure 3.

6. Interobserver variability: Acknowledge the limitation of single-reader interpretation and suggest validation through interobserver agreement in future studies.

Response: Thank you for your valuable comment. We have added a limitations section in the Discussion acknowledging the potential interpretation bias due to single-reader assessment and suggested validation through interobserver agreement in future studies.

7. Discussion clarity: Briefly explore why other expected CT indicators (mesenteric edema, ascites) were not statistically significant. The limitations section should be condensed and organized thematically for easier reading.

Response: Thank you for your comment. We have revised the Discussion to briefly explain why other expected CT indicators (e.g., mesenteric edema, ascites) were not statistically significant, noting their limited specificity and potential occurrence in non-strangulated obstruction.

We have reorganized the limitations section in the Discussion into a separate, thematically structured list for clarity and easier reading.

Other Comments

  • Correct minor typographical issues (e.g., “forth” → “fourth”), and ensure consistency in hyphenation (e.g., “preoperative” throughout).

Response: Thank you for your advice. The revised manuscript was professionally edited through MDPI Author Services to address typographical consistency.

  • Verify reference style compliance with J. Clin. Med. formatting.

Response: Thank you for your advice. The revised manuscript was professionally edited through MDPI Author Services to address formatting consistency.

  • Figures and legends are clear, but consider adding brief clinical context for educational value.

Response: Thank you for your helpful suggestion. We have revised the Figure legends to include a brief clinical context for each imaging sign to increase educational value.

  • The conclusion should stress integration of radiologic findings with clinical and intraoperative assessment, rather than relying on imaging alone.

Response: Thank you for your suggestion. We have revised the Conclusion section to emphasize integration of radiologic findings with clinical and intraoperative assessment.

Reviewer 2 Report

Comments and Suggestions for Authors

Congratulations to the authors, as the work is well designed and structured with a key question that the authors, like many others, are trying to answer. The manuscript complies with ethical guidelines and policies, fitting within the scope of the journal, and has a linguistic quality adequate for peer review. The work has several limitations, which the authors acknowledge, one of which is the sample size and evaluation by a single radiologist.

The introduction is accurate, with adequate materials and methods, as well as the figures. The results and discussion are well-structured.
However, I would like to make a series of observations:
How is a diet-related obstruction defined? This obstruction also required resection.
Is it necessary to perform a CT scan in all eventrations or incarcerated incisional hernias, as is the case with inguinofemoral hernias? Non-reducible hernias require urgent surgery, and the possibility of intestinal resection increases exponentially as time goes on.
As you mention, the time between the imaging test and surgery can indicate a progression of ischemia. Do you have this data over time? This, as you mention, is a limitation of the study.

Author Response

Congratulations to the authors, as the work is well designed and structured with a key question that the authors, like many others, are trying to answer. The manuscript complies with ethical guidelines and policies, fitting within the scope of the journal, and has a linguistic quality adequate for peer review. The work has several limitations, which the authors acknowledge, one of which is the sample size and evaluation by a single radiologist.

The introduction is accurate, with adequate materials and methods, as well as the figures. The results and discussion are well-structured.

Response: Thank you for highlighting the strengths and weaknesses regarding out manuscript. In response to your comment and Reviewer 1’s related suggestion, we have refined the patient selection criteria to exclude cases where surgery was performed more than 24 hours after contrast-enhanced CT and SSBO was not suspected at the time of surgical decision. This adjustment ensures a more homogeneous study population and minimizes potential bias related to ischemia progression over time.

As a result of this stricter criterion, the number of included cases has slightly changed. While the overall conclusions remain consistent with the original analysis, some data points have been updated accordingly. We have transparently reflected these changes in the Methods, Results, and Discussion sections.

However, I would like to make a series of observations:

How is a diet-related obstruction defined? This obstruction also required resection.

Response: Thank you for your query. In the revised manuscript, we refined the patient selection criteria to exclude cases where surgery was performed more than 24 hours after contrast-enhanced CT and SSBO was not suspected at the time of surgical decision. As a result, the previously included diet-related obstruction case was excluded from the analysis, and we are unable to provide specific data for this scenario in the current study.

Is it necessary to perform a CT scan in all eventrations or incarcerated incisional hernias, as is the case with inguinofemoral hernias? Non-reducible hernias require urgent surgery, and the possibility of intestinal resection increases exponentially as time goes on.

Response: Thank you for your comments. In Japan, access to CT in emergency departments is generally excellent, and performing contrast-enhanced CT promptly for patients with acute abdomen, including incarcerated hernias, is feasible. This approach enables assessment of bowel ischemia associated with hernia incarceration and evaluation of other intra-abdominal conditions. However, this reflects the current situation and advantages of our healthcare system, and we have not incorporated this discussion into the manuscript because the current study specifically analyzed patients who underwent preoperative contrast-enhanced CT for small bowel obstruction.

As you mention, the time between the imaging test and surgery can indicate a progression of ischemia. Do you have this data over time? This, as you mention, is a limitation of the study.

Response: Thank you for emphasizing this important point regarding progression of ischemia. In response to your comment and Reviewer 1’s related suggestion, we have refined the patient selection criteria to exclude cases where surgery was performed more than 24 hours after contrast-enhanced CT and SSBO was not suspected at the time of surgical decision. While the main conclusions remain consistent, some data points have been updated and reflected in the revised Methods, Results, and Discussion sections.

Round 2

Reviewer 2 Report

Comments and Suggestions for Authors

The authors have adequately and thoroughly addressed the observations made in the preliminary evaluation. The implemented corrections are relevant and significantly contribute to improving the scientific quality, clarity of exposition, and coherence of the work.